# A New Anti-Estrogen Discovery Platform Identifies FDA-Approved Imidazole Anti-Fungal Drugs as Bioactive Compounds against ERα Expressing Breast Cancer Cells

**DOI:** 10.3390/ijms22062915

**Published:** 2021-03-13

**Authors:** Manuela Cipolletti, Stefania Bartoloni, Claudia Busonero, Martina Parente, Stefano Leone, Filippo Acconcia

**Affiliations:** Department of Sciences, University Roma Tre, Viale Guglielmo Marconi, 446, I-00146 Rome, Italy; manuela.cipolletti@uniroma3.it (M.C.); stefania.bartoloni2@uniroma3.it (S.B.); claudia.busonero@uniroma3.it (C.B.); martina.parente@uniroma3.it (M.P.); stefano.leone@uniroma3.it (S.L.)

**Keywords:** breast cancer, estrogen receptor α, drug discovery, 17β-estradiol signaling, fenticonazole, clotrimazole, metastatic estrogen receptor α mutations

## Abstract

17β-estradiol (E2) exerts its physiological effects through the estrogen receptor α (i.e., ERα). The E2:ERα signaling allows the regulation of cell proliferation. Indeed, E2 sustains the progression of ERα positive (ERα+) breast cancers (BCs). The presence of ERα at the BC diagnosis drives their therapeutic treatment with the endocrine therapy (ET), which restrains BC progression. Nonetheless, many patients develop metastatic BCs (MBC) for which a treatment is not available. Consequently, the actual challenge is to complement the drugs available to fight ERα+ primary and MBC. Here we exploited a novel anti-estrogen discovery platform to identify new Food and Drug Administration (FDA)-approved drugs inhibiting E2:ERα signaling to cell proliferation in cellular models of primary and MBC cells. We report that the anti-fungal drugs clotrimazole (Clo) and fenticonazole (Fenti) induce ERα degradation and prevent ERα transcriptional signaling and proliferation in cells modeling primary and metastatic BC. The anti-proliferative effects of Clo and Fenti occur also in 3D cancer models (i.e., tumor spheroids) and in a synergic manner with the CDK4/CDK6 inhibitors palbociclib and abemaciclib. Therefore, Clo and Fenti behave as “anti-estrogens”-like drugs. Remarkably, the present “anti-estrogen” discovery platform represents a valuable method to rapidly identify bioactive compounds with anti-estrogenic activity.

## 1. Introduction

The sex hormone 17β-estradiol (E2) exerts pleiotropic physiological effects through its binding to the estrogen receptor α (i.e., ERα), which is a ligand-activated transcription factor. The binding of E2 to the receptor activates gene transcription. The E2-dependent association of the ERα to millions of regulatory sites on chromatin (i.e., promoters containing or not containing the estrogen-responsive-element (ERE) sequence) exerts direct control on gene expression, which is further modulated by the ability of the ERα to be recruited in different chromatin multi-protein complexes. These receptor interactors are specific transcriptional co-regulators, which contribute to the modulation of E2-induced gene transcription [1]. The E2:ERα transcriptional activity is further controlled by the extra-nuclear signaling activated by E2 through the activation of the plasma membrane located ERα (e.g., ERK/MAPK; PI3K/AKT pathways) [2,3]. The nuclear and extra-nuclear mechanism of E2:ERα molecular actions depends on the regulation of ERα concentration [4]. The ERα is a short-lived protein with a high turnover rate [4]. The modulation of the ERα intracellular levels is under the control of genetic, epigenetic (e.g., different promoter usage) and post-translational mechanisms (e.g., 26S proteasome) (for review, please see [5,6]). E2 and other receptor ligands can influence ERα cellular abundance by inducing either ERα degradation or ERα stabilization [4]. The modification of ERα intracellular levels is a critical parameter for the E2 effects. Indeed, it is finely tuned by the E2-induced extra-nuclear signaling cascades and the ERα degradation is intrinsically connected with the E2-regulated gene transcription [7,8]. The activation of these E2-induced cellular signaling circuitries determines the induction of DNA synthesis, cell cycle progression, and cell proliferation [5,6,9].

Thus, it is not surprising that E2 is involved in the growth, survival, and spreading of breast cancers (BCs). Although BC is a heterogenous disease with different molecular phenotypes, the most frequent BC is ERα positive (ERα+) at the diagnosis [10]. The presence of the ERα drives the therapeutic treatment (i.e., the endocrine therapy—ET), which uses drugs aiming to block different aspects of the E2:ERα proliferative signaling. ET drugs either reduce E2 availability (e.g., aromatase inhibitors (AIs)) or block ERα activity by directly binding to the receptor; 4OH-tamoxifen (Tam), a selective ER modulator (SERM), inactivates ERα transcriptional activity, and fulvestrant (ICI182,780—ICI), a selective ER down-regulator (SERD), eliminates ERα from BC cells [11]. In these ways, ET restrains BC progression [10]. Nonetheless, albeit its efficacy, ET has limitations as 1/3 of patients develop de novo or acquired resistance mechanisms, relapsing in metastatic BCs (MBC) [10]. Overall, a consistent fraction of ET-resistant BCs still expresses the ERα due to aberrant genetic alterations, which creates a constitutive ERα signaling loop independent by E2 stimulation [6]. Moreover, a considerable fraction of MBCs expresses ERα point mutations (e.g., Y537S ERα) that render the receptor transcriptionally hyperactive and sustain uncontrolled tumor growth [12,13,14,15,16]. Unfortunately, MBC can be only treated with limited pharmacological options as, although novel SERMs and/or SERDs or CKD4/CKD6 inhibitors are being introduced in the clinical practice, the ET drugs (e.g., Tam and ICI), alone or in combination with other chemotherapy agents, remain the mainstay of clinical treatment for women with ERα+ primary and MBC [11,17]. In particular, the novel SERDs (e.g., AZD9496 and GDC-0810) display an improved efficacy and bioavailability than ICI towards the Y537S mutated ERα, but their clinical use is still under investigation [5,18,19]. Consequently, it has become increasingly imperative to implement the BC drugs available to fight ERα+ primary and MBC in the clinic.

Nowadays drug discovery is facilitated by high-throughput techniques, which allows the identification of positive hit molecules that can undergo further development. These methods not only speed up the research process through an unbiased approach but also help to screen a high number of candidates in a very limited amount of time. In this respect, we have developed and characterized several assays to directly measure in cell lines different aspects of the E2:ERα proliferative signaling including either ERα-dependent transcriptional activity both controlled by wild type and Y537S mutant receptor, cell proliferation or ERα levels, in a high-throughput format [20,21,22].

These methods, together with the possibility to measure in vitro through non-radioactive high-content format assays the binding of virtually any kind of compounds to the recombinant purified ERα [23], represent a screening platform, which can be used to identify molecules interfering with E2:ERα signaling. Hit compounds blocking E2:ERα signaling to cell proliferation would correspond to appealing candidates working as potential “anti-estrogens”.

Here, we exploited this new anti-estrogen discovery platform to identify new Food and Drug Administration (FDA) approved drugs binding to ERα and inhibiting E2:ERα signaling to cell proliferation in cellular models of primary and MBC cells and found that the anti-fungal drugs clotrimazole (Clo) and fenticonazole (Fenti) work as potential novel “anti-estrogens” in cellular models of primary and MBC.

## 2. Results

### 2.1. The Ability of FDA-Approved Drugs to Bind ERα and Affect ERα Signaling and Cell Proliferation in Cell Lines Modelling Primary and MBC

Initially, we tested if we could find ligands of the ERα within a library of 1018 FDA-approved drugs. Therefore, we performed an in vitro fluorescence-polarization-based competitive binding assay [23]. Each compound was tested at 10^−4^ M for its ability to displace the binding of the fluorescent ligand, which mimics E2, from ERα. In all the experiments non-fluorescent E2 (10^−7^ M) and DMSO were included as positive and negative controls for the assay. Overall, we performed 12 assays in 384-well plate format. For each plate, we calculated the Z’ factor and discarded those with values < 0 [24]. We next calculated a robust Z score (Z*) [25] for each sample (Figure 1a) and then set the thresholds for a positive hit as Z* > 3.

In this list of 61 compounds potentially binding to ERα (Appendix A), we found well-known receptor ligands (e.g., estriol, resveratrol, genistein), thus suggesting the possibility to identify novel FDA-approved drugs physically associating with the ERα. Next, we filtered out from those 61 molecules 14 anti-cancer drugs (Figure 1b upper Venn diagram) and those drugs already used for treating diseases related to the endocrine system (Figure 1b lower Venn diagram) and obtained a second list containing 35 compounds (Appendix A). Finally, we manually depurated this list by 14 additional drugs (Appendix A), which were known xenoestrogens (e.g., daidzein), progestins (e.g., altrenogest), or hormones (e.g., adrenaline) and retained a final list of 21 compounds.

Next, we employed these 21 drugs in the in-cell assays described in [20,21,22] using one cell line modeling primary tumors (i.e., MCF-7 cells) and one cell line modeling MBC expressing the Y537S mutated transcriptionally hyperactive ERα (i.e., Y537S). In particular, we treated each cell line with each drug at a final treatment concentration of 2 μM for 3 and 7 days and measured both the ERα levels and cell proliferation as described in [21] as well as the wild type (wt) and Y537S mutant ERα transcriptional activity at 1 or 3 days after drug administration as described in [20,22] and searched for those molecules blocking the different aspects of ERα signaling. Figure 1c shows in red or green squares the positive hits in each assay for each cell line. As indicated by the red or by the green arrows (Figure 1c), we selected in each cell line only those compounds that were identified in at least 4 out of 6 datasets. Comparison of the 6 drugs identified in both cell lines revealed that clotrimazole (Clo) and fenticonazole (Fenti) were able to affect ERα signaling in both MCF-7 and Y537S cells while econazole and tigecycline were able to interfere with receptor functions in MCF-7 or Y537S cells, respectively.

Therefore, we next studied the impact of Clo and Fenti as novel potential “anti-estrogen” drugs.

### 2.2. Validation of Clotrimazole and Fenticonazole to Affect ERα Signaling and Cell Proliferation

The ability of Clo and Fenti to bind ERα was analyzed through an in vitro fluorescence-polarization-based competitive binding assay in a wide range of drug concentrations using E2 as a positive control. Figure 2a (and relative inset) shows that both Clo and Fenti displace the fluorescent ligand, which mimics E2, from ERα. As expected E2 almost completely displaces the fluorescent ligand (inset in Figure 2a). Calculation of the IC_50_ (i.e., K_i_) revealed that both compounds bind in vitro to the ERα in the micromolar range (i.e., ~28 µM for Clo and ~14 µM for Fenti) with Fenti having an increased affinity towards the receptor (Figure 2b).

Next, we tested if these two drugs could affect the receptor intracellular levels. As shown in Figure 3a,b, treatment of MCF-7 cells for 72 h with different doses of both Clo and Fenti induced a dose-dependent reduction in ERα intracellular levels. Moreover, Clo and Fenti administration reduced the ERα content in Y537S cells to levels comparable to those determined by the administration of the SERD ICI (Figure 3c,c’).

Dose-response curves were further performed to evaluate the impact of Clo and Fenti on the proliferation of both MCF-7 and Y537S cells. As shown in Figure 3d–g, Clo and Fenti induced a dose-dependent reduction of cell proliferation in both MCF-7 and Y537S cells. Notably, the calculated IC_50_ for Clo and Fenti at 5 days after treatment revealed that both MCF-7 and Y537S cells are more sensitive to the Fenti-induced (i.e., IC_50-Fenti_MCF-7_ = 9.94 ± 0.10 μM; IC_50-Fenti_Y537S_ = 10.64 ± 1.07 μM) than the Clo-induced (i.e., IC_50-Clo_MCF-7_ = 14.14 ± 3.02 μM; IC_50-Clo_Y537S_ = 25.17 ± 5.12 μM) anti-proliferative effect.

Overall, these data confirm that Clo and Fenti bind to the ERα in vitro and that they reduce ERα levels and cell proliferation in both MCF-7 and Y537S cells as predicted by the screening procedure.

### 2.3. Clotrimazole- and Fenticonazole-Dependent Mechanism for the Control of ERα Intracellular Concentration

To understand if Clo- and Fenti-induced reduction in ERα intracellular content occurs at the transcriptional or post-transcriptional level, MCF-7 and Y537S cells were pre-treated with the protein synthesis inhibitor cycloheximide (CHX) for 6 h before 72 h of Clo and Fenti administration. ICI was also used in these experiments as an internal control.

Results show that ICI, Clo, Fenti, and CHX reduce ERα levels, as expected. Interestingly, in either cell line both ICI, Clo, and Fenti were able to further influence the CHX-dependent reduction in ERα intracellular levels (Figure 4a,a’,b,b’).

In parallel, we also measured the ability of Clo and Fenti to influence the *ESR1* gene expression. As shown in Figure 4c,d, treatment of both MCF-7 and Y537S cells with Clo and Fenti significantly reduced the cellular levels of the mRNA encoding for ERα. As expected E2 and ICI reduced ERα mRNA levels in MCF-7 and Y537S cells, respectively. Taken together these findings suggest that Clo and Fenti control ERα content at both transcriptional and post-transcriptional levels.

### 2.4. Clotrimazole and Fenticonazole Administration Impairs ERα Transcriptional Activity in MCF-7 and Y537S Cells

Because Clo and Fenti reduce ERα intracellular content in MCF-7 and Y537S cells, we next evaluated if these two drugs could also affect receptor transcriptional activity. The ability of the ERα to control gene expression depends on its E2-induced activation. E2 administration triggers receptor phosphorylation on the S residue 118, which is a pre-requisite for full ERα transcriptional activity [26]. Thus, we text evaluated the impact of Clo and Fenti on the activation state of the ERα. To this purpose, we measured the E2-induced S118 phosphorylation status of the receptor in the presence or in the absence of E2 administration to MCF-7 cells. As expected [27], 30 min E2 administration increases the ERα S118 phosphorylation in MCF-7 cells (Figure 5a,b). Notably, pre-treatment of MCF-7 cells with Clo and Fenti completely prevented the ability of E2 to induce ERα S118 phosphorylation (Figure 5a–c).

Prompted by these results, we next measured the ERα transcriptional activity in MCF-7 cells stably expressing a construct encoding for a modified luciferase (i.e., nanoLuc) under the control of ERE-containing promoter (MCF-7NLuc) [20]. In MCF-7NLuc cells, the pre-treatment with Clo and Fenti strongly reduced the E2-dependent induction of the ERE-containing promoter activity (Figure 5d). Remarkably, Clo and Fenti also significantly reduced the basal promoter activity in the absence of E2 (Figure 5d). To further strengthen the notion that Clo and Fenti inhibit the E2-induced ERα-dependent transcriptional activity, we next employed the Y537SNLuc cells that stably express the same ERE-containing promoter activity [22], which is constitutively activated by the Y537S ERα mutant present in this CRISPR-CAS9 engineered cell line [13]. According to the previous results, we observed in Y537SNLuc cells that both Clo and Fenti reduced in a dose-dependent manner the ERE-promoter activity (Figure 5e). As expected [13,22], also ICI prevented Y537S ERα mutant ERE-promoter activity (Figure 5e).

### 2.5. Clotrimazole and Fenticonazole Administration Prevents ERα Target Gene Expression

The data reported above suggest that Clo and Fenti could reduce ERα transcriptional activity. To substantiate this evidence, we next evaluated the expression levels of 3 proteins that are the product of 3 ERα target genes (e.g., presenilin2 (pS2), cathepsin D (Cat D), and Bcl-2) [7,8,28]. MCF-7 cells were treated with Clo and Fenti in the absence and the presence of E2. Notably, E2-dependent ERα degradation is enhanced in the presence of Clo and Fenti while their administration reduces receptor intracellular levels (Figure 6a–c). In addition, Clo and Fenti prevent E2-induced accumulation of pS2 and Bcl-2 but do not prevent the E2 ability to trigger Cat D expression (Figure 6a–c). Additionally, the protein levels of pS2 and Cat D in Y537S cells [13] were analyzed. Consistent with results obtained in MCF-7 cells, administration of Clo and Fenti was able to prevent the basal level of pS2 but not of Cat D (Figure 6d,e) while as expected [13], ICI reduced the expression levels of both proteins.

Because these results partially contradict those obtained by measuring the ERα transcriptional activity by Clo and Fenti in MCF-7NLuc and Y537SNLuc cells, we next evaluated the ability of Clo and Fenti to interfere with the basal transcriptional activity of the transcriptional hyperactive Y537S ERα mutant expressed in Y537S cells [14]. Therefore we performed ERα target gene expression through RT-qPCR-based E2-sensitive gene array analysis, as previously reported [29]. Treatment of Y537S cells with Clo and Fenti revealed that 56.8% and 50.0% of the genes included in the array were significantly modulated (i.e., with a >50% variation with respect to untreated cells), respectively (Figure 6f,g, yellow slices). Remarkably, Clo reduced 60.0% of its modulated genes while Fenti reduced 73.0% of its modulated genes (Figure 6f,g, red slices).

Taken altogether, these results indicate that Clo and Fenti prevent both E2-induced ERα transcriptional activity in MCF-7 cells and reduce the transcriptional hyperactivity of the Y537S ERα mutant in Y537S cells.

### 2.6. Clotrimazole and Fenticonazole Administration Prevents E2-Induced Cell Proliferation

The activation of E2:ERα signaling results in the activation of DNA synthesis, cell cycle progression, and cell proliferation [5,6]. Therefore, Clo and Fenti impact on E2-dependent DNA synthesis and cell cycle progression was studied respectively through bromodeoxyuridine (BrdU) incorporation assay and cell cycle analysis in MCF-7 cells.

As expected, E2 triggers BrdU incorporation while pre-treatment of MCF-7 cells with both Clo and Fenti prevents the basal and E2-induced BrdU incorporation (Figure 7a,a’). Furthermore, cell cycle analysis (Figure 7b,b’) indicates that E2 treatment increases the number of cells in the S and G2 phase of the cell cycle in MCF-7 cells while Clo and Fenti lead to an accumulation of cells in the G1 phase of the cell cycle and reduces the E2-dependent entry of cells in the S and G2 phase of the cell cycle. Therefore, these findings suggest that Clo and Fenti could interfere with the ability of E2 to induce cell proliferation in MCF-7 cells.

In turn, real-time growth curve analysis was next conducted in MCF-7 cells [20,30]. Figure 7c,d show that E2 promotes MCF-7 cell proliferation, while Clo and Fenti treatment significantly reduces the basal and the E2-induced proliferation of MCF-7 cells.

Taken together, these data indicate that Clo and Fenti prevent basal and E2-induced MCF-7 cell proliferation.

### 2.7. Pre-Clinical Evaluation of Clotrimazole and Fenticonazole as Novel Drugs for Treatment of MBC

Endocrine therapy (ET) drugs display a reduced efficacy in MBCs as these relapsing diseases arise as a consequence of the prolonged patient treatment with AI and/or 4OH-Tam [11,17].

To circumvent this problem, CDK4/CDK6 inhibitors (i.e., palbociclib, ribociclib, and abemaciclib) are being introduced in the clinical practice as useful drugs to be combined with ET drugs [11,17]. Since Clo and Fenti display anti-proliferative activities in cells modeling MBC (i.e., Y537S cells), we next decided to evaluate the possibility that they could display synergistic effects with CDK4/CDK6 inhibitors. Initial experiments were performed in Y537S cells to find the palbociclib, ribociclib, and abemaciclib IC_50_ in Y537S cells. Growth curve analysis (data not shown) revealed that palbociclib (IC_50_ = 2.45 ± 0.35 µM) and abemaciclib (IC_50_ = 0.73 ± 0.12 µM) are more effective than ribociclib (IC_50_ = 18.29 ± 2.55 µM) in reducing Y537S cell proliferation. Therefore, we next treated Y537S cells with increasing concentrations of Clo, Fenti, palbocilib, and abemaciclib. Combination analyses of both Clo and Fenti with either palbociclib or abemaciclib displayed surfaces of synergy (Figure 8a,b) and significantly reduced the number of Y537S cells with respect to both untreated, Clo-, Fenti- and palbociclib- and abemaciclib-treated cells (Figure 8a–d). These data indicate that Clo and Fenti demonstrate synergism with CKD4/CDK6 inhibitors in preventing the proliferation of cell modeling MBC.

Next, we studied the anti-proliferative effects of Clo and Fenti in Y537S tumor cell spheroids, also using ICI as an internal control. Tumor spheroids were counted at time 0 (i.e., before drug administration), and at the end of the treatment (i.e., 7 days). Y537S cell spheroids grew within the experimental window (i.e., 7 days), and ICI and Clo were not able to prevent Y537S spheroid growth (Figure 9). On the contrary, Fenti reduced the number of Y537S spheroids (Figure 9). Therefore, Fenti but not Clo maintains its anti-proliferative activity also in a 3D environment.

Altogether these data suggest that Clo and Fenti could be used as an adjuvant drug in the treatment of MBC

## 3. Discussion

Breast cancer is the most commonly occurring cancer in women and the second most common cancer overall with over 2 million new cases in 2018. According to the National Cancer Institute (NIH), in 2020 BC has represented 15.3% of all new cancer cases in the U.S. with 276,480 new cases and 42,170 deaths. Although BC is a heterogeneous disease characterized by different molecular alterations, approximately 75% of BC are E2-dependent tumors characterized by high expression of the ERα. To block different aspects of E2:ERα signaling to cell proliferation ET represents a validated pharmacological strategy for the management of early and advanced ERα+ BC. Despite the ET proven efficacy, at least 1/3 of patients treated with ET drugs develop de novo or acquired resistance mechanisms that result in a relapse of the disease and in the recurrence of an MBC, for which only limited pharmacological options exist. Remarkably, most ET-resistant BC tumors remain highly addicted to constitutive E2-independent ERα signaling to cell proliferation [11].

Therefore, it is paramount, in this scenario, to approach the above-mentioned BC problems through the search and characterization of new bioactive compounds that could prevent E2:ERα signaling in primary and MBC cells to block cell proliferation. In this respect in recent years, we have shown the possibility to measure in *in-cell* assays the levels of the ERα [21], cell proliferation [21], and the ERα transcriptional activity [20,22]. The generated repertoire of assays, which can measure in a high-throughput format all aspects of E2:ERα signaling to cell proliferation in cell lines modeling both primary and MBC cells [20,21,22] together with the commercially available non-radioactive kit to evaluate in vitro competitive binding to recombinant ERα represent a new screening platform to identify compounds interfering with E2:ERα signaling to cell proliferation. Although multiple high throughput methods (e.g., in silico and in vitro ERα binding assays; screens either for drugs inhibiting receptor transcriptional activity or for anti-proliferative drugs) have been employed either to discover new ERα ligands, to identify inhibitors of ERα transcriptional activity, to induce ERα degradation or to select specific molecules that directly inhibit cell proliferation [31,32,33,34], all these assays only target one single parameter of ERα signaling at once. On the contrary, our platform has the advantage to evaluate virtually all aspects of wild type and mutant ERα signaling altogether.

Here, in the perspective to identify novel bioactive compounds prospectively usable as novel anti-BC drugs, we challenged the platform with a library of 1018 FDA-approved drugs (i.e., a drug-repurposing approach) seeking compounds directly binding to ERα and inhibiting ERα signaling to cell proliferation in cell lines modeling primary and MBC cells.

By using this approach, we have identified 3 anti-fungal drugs (i.e., econazole, clotrimazole, and fenticonazole) and 1 antibiotic (i.e., tigecycline) and further characterized the effect of Clo and Fenti because these two drugs affect ERα signaling in cellular models of primary (i.e., MCF-7) and metastatic (i.e., Y537S) BC cells. It is interesting to note that tigecycline is the election therapy for multi-drug resistant (MDR) pathogens that arise in nosocomial infections [35]. Thus, although this molecule is particularly attractive for potential MBC treatment, its use could be limited as continuous and prolonged administration to BC patients would increase the possibility to select MDR bacterial strains also resistant to tigecycline.

Validation of the effects of Clo and Fenti on the different aspects of E2:ERα signaling to cell proliferation revealed that indeed these two compounds are ERα ligands, although they associate to the receptor with different affinities, blocks ERα transcriptional signaling, reduce ERα intracellular content, and prevent basal and E2-induced cell proliferation. Remarkably, these activities occur in cells expressing the wt ERα as well as the Y537S ERα mutant. The Y537S receptor variant is transcriptionally hyperactive and represents the most common point mutant found in MBC [13]. Patients expressing this point mutation are insensitive to ET drugs as the mutated receptor assumes a 3D conformation resembling that of the E2-activated wt ERα and displays a reduced affinity towards classic and novel SERDs [16]. We did not measure the affinity of Clo and Fenti to the Y537S ERα, but, nonetheless, we reported that both drugs down-modulate its expression and inhibit cell proliferation in MBC cells, as the classic and novel SERDs do [16]. However, under no circumstances, the present work claims to define Clo and Fenti as novel SERDs. Rather, the evidence reported herein suggests that these compounds can work as “SERD-like” molecules as they inhibit the E2-induction of target genes at both the mRNA and protein level and prevent E2-induced cell proliferation.

The anti-cancer effect of Clo and Fenti was already reported [36,37] and we further demonstrate it not only in cells modeling primary tumor but also in cells modeling MBC. In addition to the ability of Clo to interfere with cancer cell metabolism, signaling, and cell cycle [37], we add ERα to the repertoire of the cellular targets of Clo. On the contrary, we show that Fenti could use ERα as one of its cellular targets. The mechanism through which Fenti inhibits cell proliferation are superimposable to those observed for Clo, at least in BC cells. Notably, BC cells appear to be more sensitive to Fenti rather than to Clo. Although future investigations are required to firmly establish the Fenti anti-cancer potential, it has to be pointed out that Fenti exerts its anti-proliferative activity with minimal toxicity in normal epithelial cells [36]. In this respect, it is important to note that the profile of the anti-cancer effects of Clo and Fenti towards different kind of tumors remains to be established but available data [36] indicate that these two FDA-approved drugs could selectively target tumor cells without significantly impacting on the cell proliferation of normal non-transformed cells, thus suggesting these two anti-fungal medications to be efficiently re-purposed as anti-tumor drugs. Nonetheless, pharmacological development of Clo and Fenti are required before establishing their clinical use. Finally, the chemical structure of these compounds could also in principle be used as leads to produce novel potential anti-cancer drugs.

Additionally, we report here that Fenti retains its anti-proliferative activity in 3D models of MBC (i.e., tumor cell spheroids). More importantly, Clo and Fenti synergize with the CDK4/CDK6 inhibitors palbociclib and abemaciclib that are being used in the clinics as an adjuvant for MBC management. Therefore, altogether this evidence suggests that Clo and Fenti could work as novel FDA-approved compounds active against primary and metastatic BCs.

In conclusion, we demonstrate here that by exploiting our novel screening platform it is possible to identify compounds like econazole, clotrimazole, fenticonazole, and tigecycline working as novel “anti-estrogens” preventing E2:ERα signaling to BC cell proliferation. Of course, the same platform can be further exploited to identify novel pathways affecting E2:ERα signaling to BC cell proliferation by applying siRNA or CRISPR/CAS9 libraries directed against the specific protein of choice, thus providing the opportunity to pinpoint novel druggable proteins and/or cellular pathways potentially usable in BC treatments.

## 4. Materials and Methods

### 4.1. Cell Culture and Reagents

17β-estradiol (E2), DMEM (with and without phenol red), and fetal calf serum were purchased from Sigma-Aldrich (St. Louis, MO, USA). Bradford protein assay kit, as well as anti-mouse and anti-rabbit secondary antibodies, were obtained from Bio-Rad (Hercules, CA, USA). Antibodies against ERα (H-C20, rabbit), Bcl-2 (C2, mouse), cathepsin D (H75, rabbit), pS2 (FL-84, rabbit) were obtained from Santa Cruz Biotechnology (Santa Cruz, CA, USA); anti-phospho ERα (Ser118, mouse) antibody was obtained from Cell Signaling; anti-vinculin (mouse) antibody was purchased from Sigma-Aldrich (St. Louis, MO, USA). Chemiluminescence reagent for Western blotting was obtained from BioRad Laboratories (Hercules, CA, USA). Fulvestrant (i.e., ICI182,780) was purchased by Tocris (Bristol, UK), Cycloheximide (CHX) was purchased from Sigma-Aldrich (St. Louis, MO, USA). FDA-approved drug library as well as palbociclib, ribociclib, and abemaciclib were purchased by Selleck Chemicals (Houston, TX, USA). PolarScreen™ ERα Competitor Assay Kit, Green (A15882) was acquired from Thermo Scientific (Waltham, MA, USA). All the other products were from Sigma-Aldrich. Analytical- or reagent-grade products were used without further purification. The identities of all the used cell lines were verified by STR analysis (BMR Genomics, Padova, Italy).

### 4.2. In Vitro ERα Binding Assay

A fluorescence polarization (FP) assay was used to measure the binding affinity of 1,018 FDA-approved compounds and 17β-estradiol (E2) for recombinant ERα in vitro. The FP assay was performed using a PolarScreen™ ERα Competitor Assay Kit, Green (A15882, Thermo Scientific) as previously reported [29].

### 4.3. In-Cell Western Blotting

In-cell Western blot was used to measure ERα levels in MCF-7 and Y537S cell lines. The experiments were carried on using the protocol previously described [21]. The cells were treated with the selected compounds at a concentration of 2 µM for 3 and 7 days. ICI (1 µM) was used as a control for ERα degradation.

### 4.4. In-Cell Propidium Iodide (PI) Staining

In-cell PI staining was used to measure DNA content in MCF-7 and Y537S cell lines. The experiments were carried on using the protocol previously described [21]. The cells were treated with the selected compounds at a concentration of 2 µM for 3 and 7 days. Taxol (1 µM) was used as control for cell proliferation.

### 4.5. Real-Time Measurement of ERα Transcriptional Activity

MCF-7 and Y537S cells were stably transfected with a plasmid containing an ERE-nanoluciferase (NLuc)-PEST reporter gene and real-time measurement of NLuc-PEST expression (i.e., ERα transcriptional activity) was performed as described [20,22].

### 4.6. Cell Manipulation for Western Blotting Analyses

Cells were grown in DMEM with phenol red plus 10% fetal calf serum for 24 h and then treated with fenticonazole (Fenti), clotrimazole (Clo), or fulvestrant (ICI) at the indicated doses for the indicated periods. Before E2 stimulation, cells were grown in DMEM without phenol red plus 10% charcoal-stripped fetal calf serum for 24 h; Fenti, Clo, and cycloheximide (CHX) were added before E2 administration. After treatment, cells were lysed in Yoss Yarden (YY) buffer (50 mM Hepes (pH 7.5), 10% glycerol, 150 mM NaCl, 1% Triton X-100, 1 mM EDTA and 1 mM EGTA) plus protease and phosphatase inhibitors. Western blot analysis was performed by loading 20–30 µg of protein on SDS-gels. Gels were run, and the proteins were transferred to nitrocellulose membranes with a Turbo-Blot semidry transfer apparatus from Bio-Rad (Hercules, CA, USA). Immunoblotting was carried out by incubating the membranes with 5% milk or bovine serum albumin (60 min), followed by incubation overnight (o.n.) with the indicated antibodies. Secondary antibody incubation was continued for an additional 60 min. Bands were detected using a Chemidoc apparatus from Bio-Rad (Hercules, CA, USA).

### 4.7. Cell Proliferation and Cell Cycle Assays

For growth curves and drug synergy studies the xCELLigence DP system ACEA Biosciences, Inc. (San Diego, CA, USA) Multi-E-Plate station was used to measure the time-dependent response to the indicated drugs by real-time cell analysis (RTCA), as previously reported [20,29,30]. Briefly, the number of cells (i.e., normalized cell index) is directly proportional to the measured electric impedance of the cells on the well surface. Cells were seeded in E-Plates 96 in the growing medium. After overnight monitoring of growth once every 15 min, drugs were added. Cells remained in the medium until the end of the experiment. Cellular responses were then recorded once every 15 min for a total time of 5 days. Next, the synergy index was calculated with Combenefit freeware software [38]. For cell cycle analysis, after each treatment, 1 × 10^6^ cells were washed twice with PBS, fixed dropwise with ice-cold ethanol (70%), and rehydrated with PBS. DNA staining was performed by incubating cells for 30 min at 37 °C in PBS containing 0.18 mg/mL propidium iodide (PI) and 0.4 mg/mL DNase-free RNase (type 1-A). Samples were acquired with a CytoFlex Flow Cytometer (Beckman Coulter) equipped with 488 nm and 635 nm laser sources. Cell cycle analysis was performed using CytExpert v.2.4 software (Beckman Coulter). Doublet discrimination was performed by an electronic gate on FL2-Area vs. FL2-Height parameters. For cell cycle analysis, Nicoletti’s protocol was followed [39]. Briefly, the cell pellet was resuspended in 500 µL of PBS, fixed by adding 4.5 mL of 70% cold ethanol, washed twice, and resuspended in 500 µL of PBS + 500 µl of DNA extraction buffer (0.19 M Na_2_HPO_4_, 0.004% Triton X-100, pH 7.8). Cells were incubated for 5 min at room temperature. Pellet was resuspended in 1 mL of DNA staining solution (20 µg of propidium iodide, 0.2 mg of RNaseA, in PBS) and incubated once again for 30 min at room temperature. Finally, 20,000 total events on a linear scale were acquired and the percentage of each cell cycle phase was calculated by a proper electronic marker.

### 4.8. RNA Isolation and qPCR Analysis

The sequences for gene-specific forward and reverse primers were designed using the OligoPerfect Designer software program (Invitrogen, Carlsbad, CA, USA). The following primers were used: human ERα: 5 0-GTGCCTGGCTAGAGATCCTG-3 0 (forward) and 5 0-AGAGACTTCAGGGTGCTGGA-3 0 (reverse) and human GAPDH: 5 0-CGAGATCCCTCCAAAATCAA-3 0 (forward) and 5 0-TGTGGTCAT GAGTCCTTCCA-3 0 (reverse). Total RNA was extracted from cells using TRIzol reagent (Invitrogen, Carlsbad, CA, USA) according to the manufacturer’s instructions. To determine gene expression levels, cDNA synthesis and qPCR were performed using the GoTaq 2-step RT-qPCR system (Promega, Madison, MA, USA) in an ABI Prism 7900 HT Sequence Detection System (Applied Biosystems, Foster City, CA, USA) according to the manufacturer’s instructions. Experiments were performed in triplicate. The gene expression was normalized for GAPDH mRNA levels. RT-qPCR-based gene array analysis of ERα target gene expression was determined using the PrimePCR Estrogen receptor signaling (SAB Target List) H96 panel (Bio-Rad Laboratories, Hercules, CA, USA) according to the manufacturer’s instructions. Gene expression was normalized to GAPDH mRNA levels.

### 4.9. Bromodeoxyuridine Incorporation Assay

Bromodeoxyuridine (BrdU) was added to the medium in the last 30 min of growth, and the cells were then fixed and permeabilized. Histones were dissociated with 2 M HCl as previously described [40]. BrdU-positive cells were detected with anti-BrdU primary antibody diluted 1:100 (DAKO; Santa Clara, CA, USA) and Alexa488-conjugated anti-mouse antibody diluted 1:100 (Thermo Fisher Scientific; Waltham, MA, USA). Both antibodies were incubated with the cells for 1 h at room temperature in the dark. BrdU fluorescence was measured using a CytoFlex flow cytometer, and S-phase analysis was performed with CytExpert v 2.3 software (Beckman Coulter, Brea, CA, USA). All samples were counterstained with propidium iodide (PI) for DNA/BrdU biparametric analysis.

### 4.10. Tumor Spheroid Formation

Tumor spheroid formation was performed as previously reported [30]. Briefly, Y537S cells were seeded (10,000 cells/well) in ultra-low attachment surface 24-well-plates (Sigma-Aldrich) with 1 mL/well in growing condition for 48 h. Next, using an optical microscope, pictures had been taken for each well in untreated conditions (i.e., time 0). Then, cells were treated in quadruplicate with the indicated compounds and with vehicle (DMSO). After 48 h, the cell culture medium was changed using a 70 µm nylon sterile cell strainer for each condition to maintain spheroids with a diameter greater than 70 µm and to remove dead cells and spheroids with a diameter smaller than 70 µm. Contemporarily, the treatment was repeated. Seven days post initial drug administration, pictures had been taken for each well. The number of spheroids has been quantitated using the freeware software Image J.

### 4.11. Statistical Analysis

Statistical analysis was performed using the InStat version 8 software system (GraphPad Software Inc., San Diego, CA, USA). Densitometric analyses were performed using the freeware software Image J by quantifying the band intensity of the protein of interest with respect to the relative loading control band (i.e., vinculin) intensity. The *p* values and the used statistical test are given in figure captions.

## Figures and Tables

**Figure 1 ijms-22-02915-f001:**
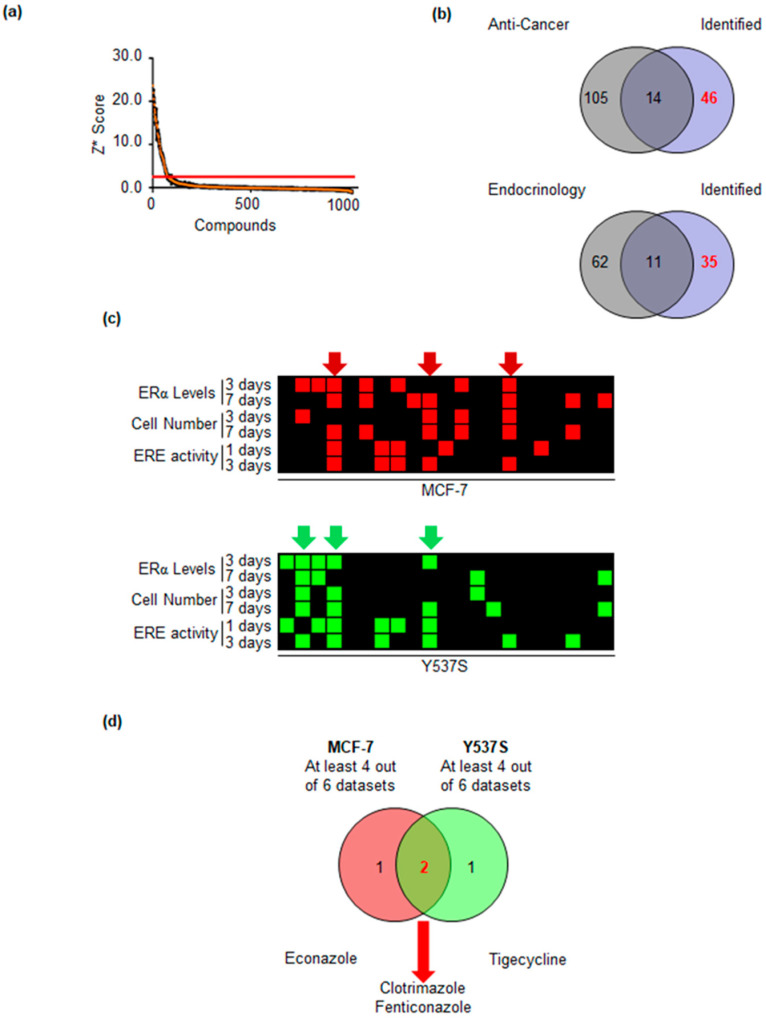
Identification of FDA-approved drugs binding to the ERα. (**a**) Z* for each of the 1018 compounds in the FDA-approved drug library toward in vitro ERα competitive binding assays (**b**) Venn diagrams depicting the procedure to filter out from the initial list the anti-cancer and endocrinological drugs. (**c**) Heatmap depicting the positive hits of the 21 selected drugs towards their ability to modulate ERα levels, transcriptional activity, and cell proliferation in both MCF-7 (upper panel) and Y537S (lower panel) cells. (**d**) Venn diagram depicting the identified drugs.

**Figure 2 ijms-22-02915-f002:**
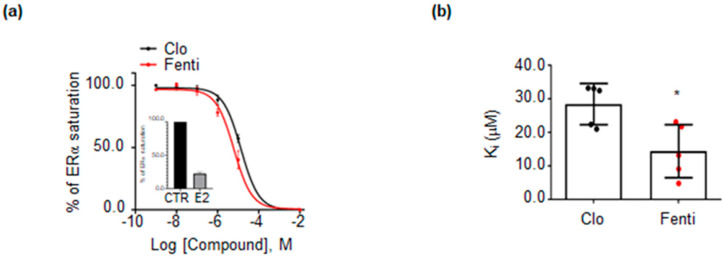
**Validation of in vitro binding of clotrimazole and fenticonazole to the ERα**. (**a**) In vitro ERα competitive binding assays for clotrimazole (Clo–black) and fenticonazole (Fenti-red) were performed at different doses of the compounds and using a florescent E2 as a tracer. Inset shows the in vitro ERα competitive binding assays for E2 (10^−7^ M). (**b**) Relative inhibitor concentration 50 (IC_50_–µM, i.e., K_i_). The experiment was performed twice in quintuplicate. * indicates significant differences, calculated with the Student t-test, with respect to Clo, * *p*-value < 0.05.

**Figure 3 ijms-22-02915-f003:**
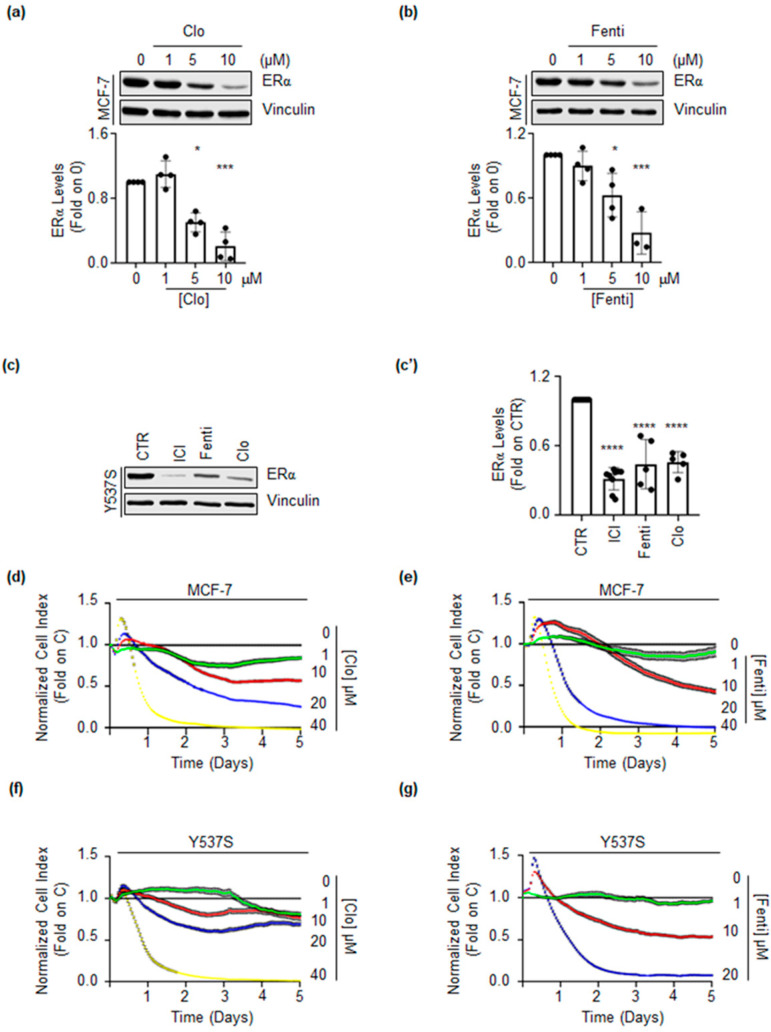
Clotrimazole and fenticonazole effect on ERα levels and cell proliferation. (**a**,**b**) Western blotting and relative densitometric analyses of ERα cellular levels in MCF-7 cells treated with different doses of clotrimazole (Clo) (**a**) and fenticonazole (Fenti) (**b**) for 72 h. The loading control was done by evaluating vinculin expression in the same filter. The experiment was performed in triplicate. Significant differences with respect to 0 sample are calculate by Student t-test and indicated by * (*p*-value < 0.05), *** (*p*-value < 0.001). Western blotting (**c**) and relative densitometric analyses (**c’**) of ERα cellular levels in Y537S cells treated with clotrimazole (Clo 10 µM) and fenticonazole (Fenti 10 µM) for 72 h. The experiment was performed in triplicate. Significant differences with respect to 0 samples are calculated by the Student t-test and indicated by **** (*p*-value < 0.0001). Growth curve analyses in MCF-7 (**d**,**e**) or in Y537S (**f**,**g**) were performed as indicated in the material and method section for 5 days with the indicated doses (color lines) of clotrimazole (Clo) (**d**,**f**) or fenticonazole (Fenti) (**e**,**g**). Experiments were done twice in quadruplicate.

**Figure 4 ijms-22-02915-f004:**
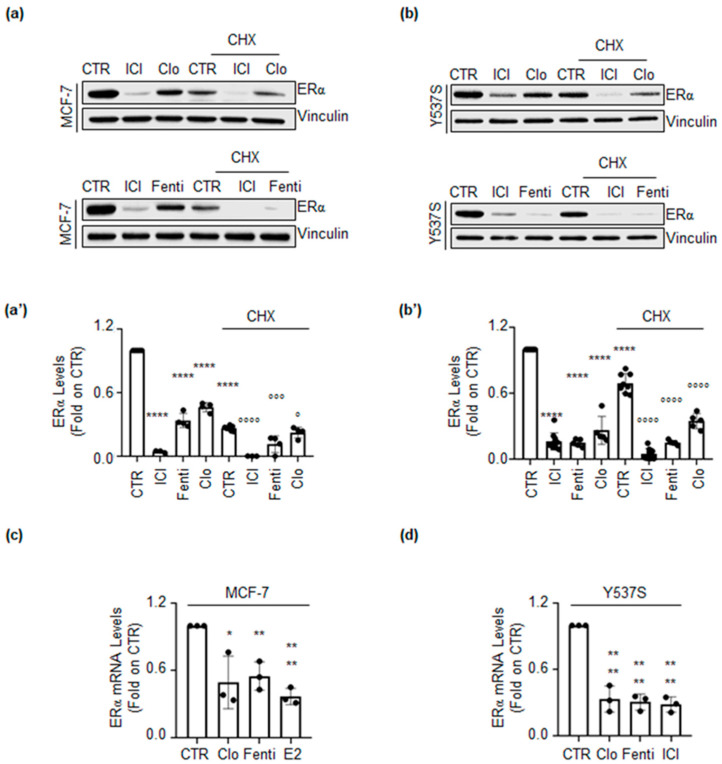
Transcriptional and post-transcriptional regulation of ERα levels by clotrimazole and fenticonazole. Western blotting and relative densitometric analysis of ERα levels in MCF-7 (**a**) and (**a’**) or in Y537S (**b**) and (**b’**) cells pre-treated with cycloheximide (CHX 1 µg/mL for MCF-7 cells and 0.1 µg/mL for Y537S cells) for 6 h and then treated with fulvestrant (ICI 100nM), clotrimazole (Clo 10 µM) and fenticonazole (Fenti 10 µM) for 72 h. The loading control was done by evaluating vinculin expression in the same filter. Panels show representative blots of three independent experiments. Significant differences with respect to the CTR sample are calculated by Student t-test and indicated by **** (*p*-value < 0.0001). Significant differences with respect to CHX sample are calculated by Student t-test and indicated by (° *p*-value < 0.05), °°° (*p*-value < 0.001) and °°°° (*p*-value < 0.0001). (**c**) RT-qPCR analysis of ERα mRNA levels in MCF-7 (c) and Y537S (**d**) cells treated with Clo or Fenti (10 µM) or E2 (1 nM) and ICI (1 µM) for 72 h. ERα mRNA expression was normalized to the GAPDH mRNA expression. The experiments were performed in triplicate. Significant differences with respect to CTR sample are calculated by Student t-test and indicated by * (*p*-value < 0.05), ** (*p*-value < 0.01) and **** (*p*-value < 0.0001).

**Figure 5 ijms-22-02915-f005:**
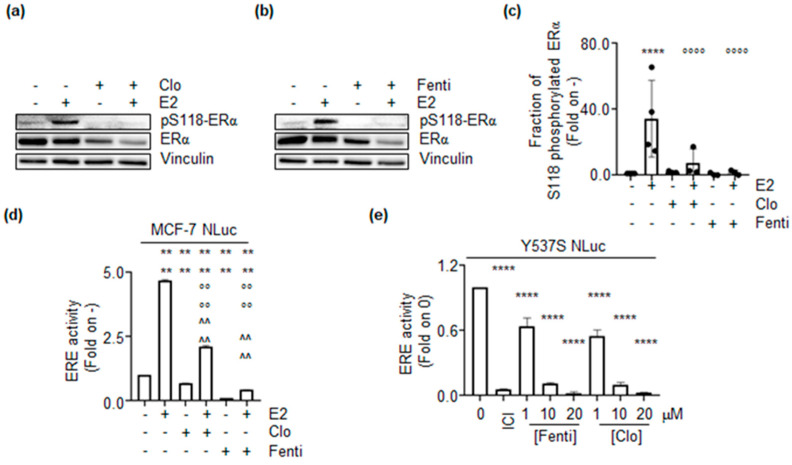
The impact of clotrimazole and fenticonazole on ERα transcriptional activation. Western blotting and analysis of the ERα and the ERα phosphorylation status on S residue 118 (pS118) induced by 17β-estradiol (E2 1 nM–30 min) in MCF-7 cells pre-treated with clotrimazole (Clo 10 µM) (**a**) and fenticonazole (Fenti 10 µM) (**b**) for 72 h. (**c**) Densitometric analysis is relative to panel (**a**,**b**). The loading control was done by evaluating vinculin expression in the same filter. Panels show representative blots of three independent experiments. Significant differences with respect to-sample are calculated by Student t-test and indicated by **** (*p*-value < 0.0001). Significant differences with respect to the E2 sample are calculated by Student t-test and indicated by °°°° (*p*-value < 0.0001). (**d**) Estrogen response element promoter activity in MCF-7 ERENLuc cells pre-treated with clotrimazole (Clo 10 µM) and fenticonazole (Fenti 10 µM) for 72 h and then treated with 17β-estradiol (E2 1nM) for an additional 24 h. (**e**) Estrogen response element promoter activity in Y537S ERENLuc cells treated with fulvestrant (ICI-100nM), clotrimazole (Clo), and fenticonazole (Fenti) at the indicated doses for 72 h. The experiments were performed twice in duplicate. Significant differences with respect to untreated (i.e., -,-,- or 0) sample are calculated by Student t-test and indicated by **** (*p*-value < 0.0001). Significant differences with respect to the E2 sample are calculated by Student t-test and indicated by °°°° (*p*-value < 0.0001). Significant differences with respect to Clo- or Fenti-alone treated sample are calculated by Student t-test and indicated by ^^^^ (*p*-value < 0.0001).

**Figure 6 ijms-22-02915-f006:**
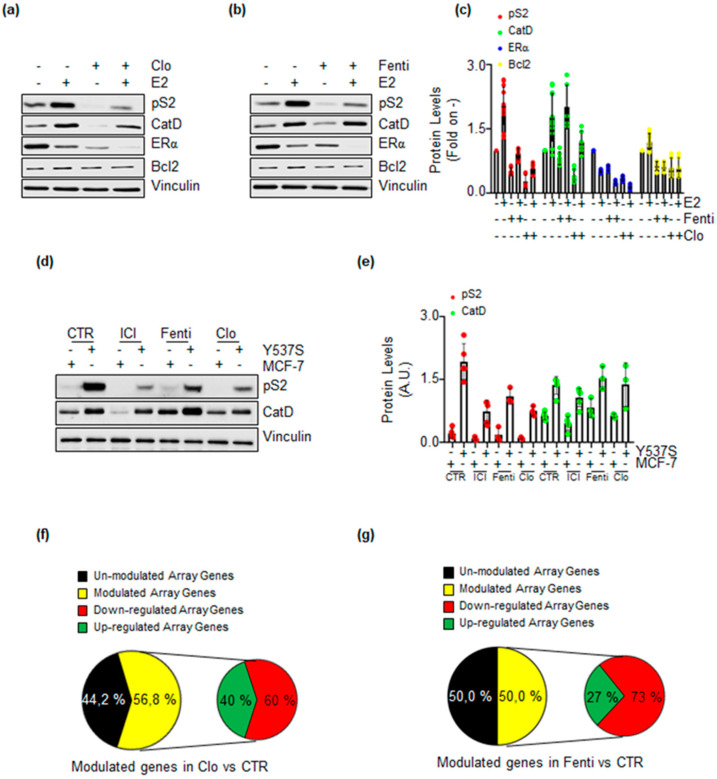
The impact of clotrimazole and fenticonazole on ERα-dependent gene expression. Western blotting analysis of ERα, presenilin 2 (pS2), cathepsin D (Cat D), and Bcl-2 expression in MCF-7 cells pre-treated with clotrimazole (Clo 10 µM) (**a**) and fenticonazole (Fenti 10 µM) (**b**) for 72 h. (**c**) Densitometric analysis relative to panel (**a**,**b**). The loading control was done by evaluating vinculin expression in the same filter. Panels show representative blots of three independent experiments. Western blotting (**d**) and relative densitometric analyses (**e**) of pS2 and Cat D protein levels in Y537S cells compared to MCF-7 cells. Cells were treated with clotrimazole (Clo 10 µM) and fenticonazole (Fenti 10 µM) for 72 h. The loading control was done by evaluating vinculin expression on the same filter. Panels show representative blots of three independent experiments. Pie diagrams representing the percentage of the array genes modulated in Y537S cells treated for 72 h clotrimazole (Clo 10 µM) (**f**) and fenticonazole (Fenti 10 µM) (**g**).

**Figure 7 ijms-22-02915-f007:**
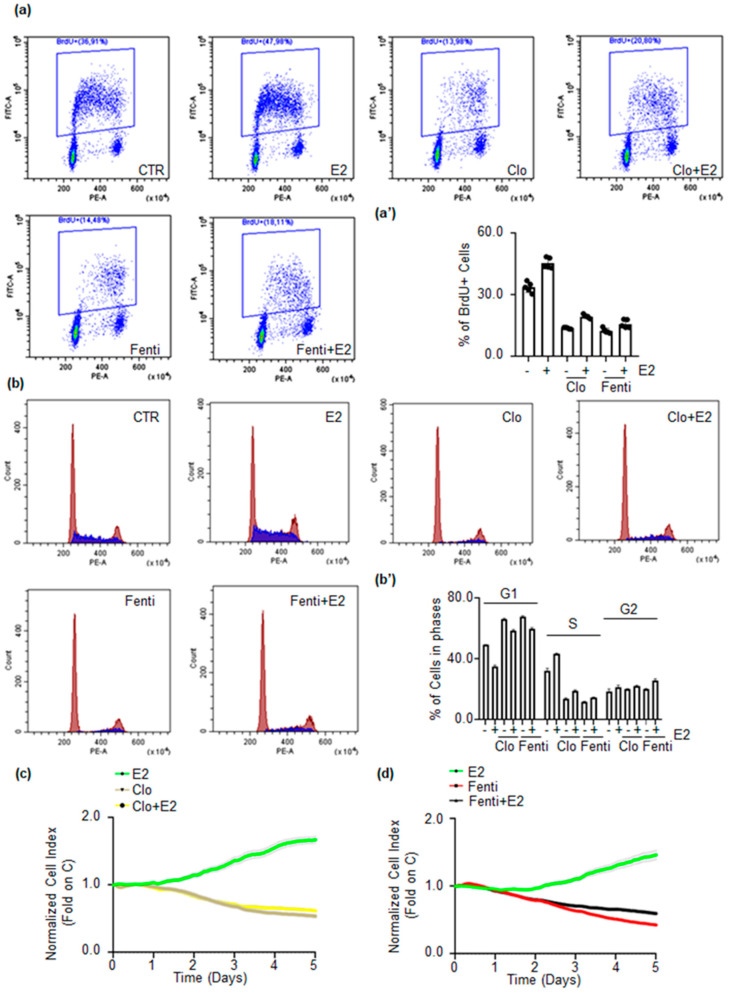
The impact of clotrimazole and fenticonazole on E2-induced DNA synthesis and cell proliferation. (**a**,**a’**) Bromodeoxyuridine (BrdU) incorporation assay and (**b**,**b’**) cell cycle analysis in MCF-7 cells treated with 17β-estradiol (E2 1 nM–24 h) after 72 h pre-treatment with clotrimazole (Clo 10 µM) and fenticonazole (Fenti 10 µM). The experiments have been performed twice in duplicate. Real-time growth curves in MCF-7 cells treated with clotrimazole (Clo 10 µM) (**c**) and fenticonazole (Fenti 10 µM) (**d**) in the absence and the presence of 17β-estradiol (E2 1 nM) and with E2 alone (1 nM) for the indicated time points. The graphs show Clo, Fenti, and E2 effect on the normalized cell index (i.e., cell number), which is detected with the xCelligence DP device and calculated at each time point with respect to the control sample. Each sample was measured in a quadruplicate. For details, please see the material and methods section.

**Figure 8 ijms-22-02915-f008:**
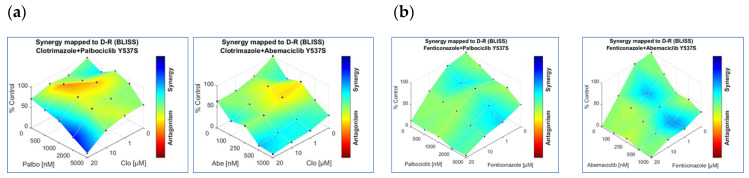
Clotrimazole and fenticonazole synergic effects with palbociclib and abemaciclib in Y537S cells. Synergy map of 5 days treated Y537S cells with different doses of clotrimazole (Clo) and palbociclib (palbo) or abemaciclib (Abe) (**a**), left and right panels, respectively) or of fenticonazole and palbociclib (palbo) or abemaciclib (Abe) (**b**), left and right panels, respectively). Growth curves in Y537S cells showing the synergic effect of each combination of compounds with selected doses are shown in (**c**–**f**). For details, please see the material and methods section.

**Figure 9 ijms-22-02915-f009:**
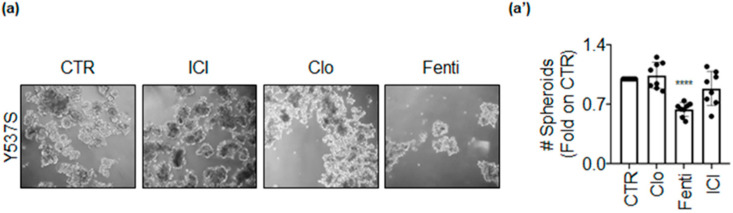
Clotrimazole and fenticonazole effects in Y537S-derived tumor spheroids. Pictures (**a**) and quantitation (**a’**) of tumor spheroids generated in Y537S cells treated at time 0 with ICI182,780 (ICI–1 µM), clotrimazole (Clo–20 µM), fenticonazole (Fenti–20 µM) or left untreated (CTR) for 7 days. Experiments were performed twice in quadruplicate. Significant differences with respect to the CTR sample were determined by unpaired two-tailed Student’s t-test: **** *p* < 0.0001.

## Data Availability

The data presented in this study are available on request from the corresponding author.

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
