# Peer review of "A New Anti-Estrogen Discovery Platform Identifies FDA-Approved Imidazole Anti-Fungal Drugs as Bioactive Compounds against ERα Expressing Breast Cancer Cells"

_ijms, 2021, doi:10.3390/ijms22062915_

Round 1
Reviewer 1 Report
Acconcia et al reported a novel assay to high throughput screen a small set of FDA approved drugs as bioactive ligands inhibiting E2:ERα signaling to cell proliferation in cellular models of primary and MBC cells. The authors found that the antifungal drugs clotrimazole and fenticonazole induce ERα degradation and prevent ERα transcriptional signaling and proliferation in primary and metastatic BC cellls. Interestingly, the authors showed that the anti-proliferative effects of clotrimazole and fenticonazole present in tumor spheroids in a synergic manner with the spalbociclib and abemaciclib which indicates that these drugs could be considered as novel anti-estrogens drugs. This is very interesting findings, the research study is well designed and provide a novel platform for screening of potential anti-estrogenic drugs. I recommend the publication of this paper after considering the following comments:
- the introduction part lack from citation of relevant studies. The authors must carefully check the introduction part.
- additionally, the authors should discuss the previously known screening assays for anti-estrogenic compounds and what is the advantages of the newly established assay?
- the authors should also discuss in the intro part the previously known (potential) anti-estrogenic drugs and why the scientific community seeks for novel and more potential anti-estrogenic drugs?
- the presented study showed clearly that the clotrimazole and fenticonazole are potential anti-estrogenic drugs with anti-proliferative effects on tumors, it would be interesting if the authors include in their discussion part what could be the selectivity of these drugs as anticancers.
- I think it would be better to include a positive control in the validation of in vitro binding of clotrimazole and fenticonazole to the ERα.
- it would be also interesting if the authors do a computational study for the clotrimazole and fenticonazole to show their affinity and binding activity to the ERα.
- the authors should also include/discuss the chemistry of these drugs. It is clear that both drugs are mainly hydrophobic and their binding to the cavity would mainly based on a hydrogen acceptor position on the scaffold. this could help in concluding or suggesting such drugs as potential scaffold for further modifications or SAR study to improve their anti-estrogenic activity?
Author Response
Reviewer #1
Acconcia et al reported a novel assay to high throughput screen a small set of FDA approved drugs as bioactive ligands inhibiting E2:ERα signaling to cell proliferation in cellular models of primary and MBC cells. The authors found that the antifungal drugs clotrimazole and fenticonazole induce ERα degradation and prevent ERα transcriptional signaling and proliferation in primary and metastatic BC cellls. Interestingly, the authors showed that the anti-proliferative effects of clotrimazole and fenticonazole present in tumor spheroids in a synergic manner with the spalbociclib and abemaciclib which indicates that these drugs could be considered as novel anti-estrogens drugs. This is very interesting findings, the research study is well designed and provide a novel platform for screening of potential anti-estrogenic drugs. I recommend the publication of this paper after considering the following comments:
Author Response: We thank the Reviewer for these comments.
- the introduction part lack from citation of relevant studies. The authors must carefully check the introduction part.
Author Response: We thank the Reviewer for these comments and now cited the relevant studies regarding the concepts presented in the discussion part.
- additionally, the authors should discuss the previously known screening assays for anti-estrogenic compounds and what is the advantages of the newly established assay?
Author Response: We have now discussed them in the discussion sections.
- the authors should also discuss in the intro part the previously known (potential) anti-estrogenic drugs and why the scientific community seeks for novel and more potential anti-estrogenic drugs?
Author Response: We already introduced the fact that despite its proven efficacy the endocrine therapy drugs possess serious side-effects. As per this request we also introduced the efficacy of the novel SERDs AZD9496 and GDC-0810 and further pointed out that they clinical use is still under investigation. We have also added the relative reference to the original works. Finally, we concluded, as already written in the original introduction section that Consequently, it has become increasingly imperative to implement the BC drugs available to fight ERα+ primary and MBC in the clinic.
- the presented study showed clearly that the clotrimazole and fenticonazole are potential anti-estrogenic drugs with anti-proliferative effects on tumors, it would be interesting if the authors include in their discussion part what could be the selectivity of these drugs as anticancers.
Author Response: We added as per this request a short paragraph in the discussion section.
- I think it would be better to include a positive control in the validation of in vitro binding of clotrimazole and fenticonazole to the ERα.
Author Response: We thank the Reviewer for raising this point. In each in vitro binding assay we always use saturating concentration of E2 as positive control. We have now added this control as an inset in figure 2a.
- it would be also interesting if the authors do a computational study for the clotrimazole and fenticonazole to show their affinity and binding activity to the ERα.
Author Response: We thank the Reviewer for this suggestion. We agree that performing a computational study to address the biochemical issues regarding the chemical mechanisms with which Clo and Fenti bind to the surface of the ERα is very interesting. However, this experiment is out of the scope of the present work, which was aimed to demonstrate that by measuring different aspects of the ERα signaling (i.e., by allowing the functioning of the screening platform) it was possible to identify in a rapid and cost-effective manner bioactive compounds with anti-estrogen activity. However, continuation of the present work is indeed focused exactly to the experiment suggested by this Reviewer.
- the authors should also include/discuss the chemistry of these drugs. It is clear that both drugs are mainly hydrophobic and their binding to the cavity would mainly based on a hydrogen acceptor position on the scaffold. this could help in concluding or suggesting such drugs as potential scaffold for further modifications or SAR study to improve their anti-estrogenic activity?
Author Response: We thank the Reviewer for this suggestion. As pointed above, we agree with this Reviewer but the present request is out of the scope of the work and, more importantly, is the focus of ongoing investigations in our laboratory.
Reviewer 2 Report
Overall, the authors designed and described the study very well. The results are clearly presented. Some minor spelling errors might be double checked by the editor.
Results part: Figure 4: for the detection of the ER RNA level, a positive control would be appreciated
Discussion: the discussion misses a broader comparison with published data and alternative treatment options for breast cancer. In addition, the author should take into account that they do have indeed very promising results. Nevertheless, they showed antitumor activity in vitro as a proof of principle. Although these compounds are already FDA proven, the therapeutic area is a completely different one. The authors should give an outlook of a possible drug development path for those compounds in oncology.
Author Response
Reviewer #2
Overall, the authors designed and described the study very well. The results are clearly presented. Some minor spelling errors might be double checked by the editor.
Author Response: We thank the Reviewer for these comments.
Results part: Figure 4: for the detection of the ER RNA level, a positive control would be appreciated
Author Response: We added the effect of E2 on the levels of ESR1 in MCF-7 and the effect of ICI on the levels of ESR1 in Y537S cells.
Discussion: the discussion misses a broader comparison with published data and alternative treatment options for breast cancer. In addition, the author should take into account that they do have indeed very promising results. Nevertheless, they showed antitumor activity in vitro as a proof of principle. Although these compounds are already FDA proven, the therapeutic area is a completely different one. The authors should give an outlook of a possible drug development path for those compounds in oncology.
Author Response: We thank the Reviewer for this comment. Regarding the discussion of the broader context of the published data with the alternative treatments for breast cancer, we believe it will put the discussion out of focus as it deserves a Review Article rather that a part of the discussion in a Research Article. In this respect we have done this in Busonero et al., 2019. Nonetheless, as per Reviewer #1 request we have better clarified in the introduction section the difficulties in finding novel anti-BC drugs targeting (or not) the ERalpha signaling. Regarding the clinical development of clotrimazole and fenticonazole, we agree with this Reviewer and added the following part to the discussion section: ‘In this respect, it is important to note that the profile of the anti-cancer effects of Clo and Fenti towards different kind of tumors still remain to be established but available data indicate that these two FDA-approved drugs could selectively target tumor cells without significantly impacting on the cell proliferation of normal non-transformed cells, thus suggesting these two anti-fungal medications to be efficiently re-purposed as anti-tumor drugs. Nonetheless, pharmacological development of Clo and Fenti are required before establishing their clinical use. Finally, the chemical structure of this compounds could also in principle be used as lead to produce novel potential anti-cancer drugs.’
Round 2
Reviewer 1 Report
The authors have significantly improved their MS. They have also responded to all points that have been raised in the reviewing process, Thanks!! Accordingly, I recommend the publication of this interesting study.